# Longitudinal viral shedding and antibody response characteristics of men with acute infection of monkeypox virus: a prospective cohort study

Yang Yang [1,2,3,7] ✉, Shiyu Niu[1,2,3,7], Chenguang Shen [4,7], Liuqing Yang[1,2,7], Shuo Song [1,2,7], Yun Peng[1,2], Yifan Xu[1,2], Liping Guo[1,2], Liang Shen[5], Zhonghui Liao[6], Jiexiang Liu[1,2], Shengjie Zhang [1,2], Yanxin Cui[6], Jiayin Chen[3], Si Chen[1,2], Ting Huang[1,2], Fuxiang Wang [1,2,3] ✉, Hongzhou Lu[1,2,3] ✉ & Yingxia Liu [1,2] ✉

Understanding of infection dynamics is important for public health measures against monkeypox virus (MPXV) infection. Herein, samples from multiple body sites and environmental fomites of 77 acute MPXV infections (HIV co-infection: N = 42) were collected every two to three days and used for detection of MPXV DNA, surface protein specific antibodies and neutralizing titers. Skin lesions show 100% positivity rate of MPXV DNA, followed by rectum (88.16%), saliva (83.78%) and oropharynx (78.95%). Positivity rate of oropharynx decreases rapidly after 7 days post symptom onset (d.p.o), while the rectum and saliva maintain a positivity rate similar to skin lesions. Viral dynamics are similar among skin lesions, saliva and oropharynx, with a peak at about 6 d.p.o. In contrast, viral levels in the rectum peak at the beginning of symptom onset and decrease rapidly thereafter. 52.66% of environmental fomite swabs are positive for MPXV DNA, with highest positivity rate (69.89%) from air-conditioning air outlets. High seropositivity against A29L (100%) and H3L (94.74%) are detected, while a correlation between IgG endpoint titers and neutralizing titers is only found for A29L. Most indexes are similar between HIV and Non-HIV participants, while HIV and rectitis are associated with higher viral loads in rectum.

Mpox (formerly known as monkeypox) is a zoonotic illness caused by the monkeypox virus (MPXV), and the first case of human Mpox case was reported in 1970 in the Democratic Republic of the Congo[1]. Thereafter, sporadic cases caused by two distinct clades were confined to Central Africa (Clade 1) and West Africa (Clade 2) for decades until international transmission was first detected in the United States in 2003[2]. Since May 2022, another new lineage MPXV (Clade 2b) derived from West African clade with certain genetic changes including C > T or G > A mutations mediated by apolipoprotein B messenger RNA (mRNA) editing catalytic polypeptide-like 3 (APO-BEC3) enzymes caused a widespread epidemic worldwide, leading to the declaration of the Mpox outbreak a Public Health Emergency of International Concern (PHEIC) by World Health Organization (WHO) on July 23, 2022[3,4]. The first imported Mpox case into mainland China

A full list of affiliations appears at the end of the paper. ✉e-mail: young@mail.sustech.edu.cn; 13927486077@163.com; luhongzhou@szsy.sustech.edu.cn; yingxialiu@hotmail.com

was found in September 2022, and no subsequent transmission occurred with the rapid diagnosis and isolation[5]. However, Mpox has been endemic in several cities of mainland China since June 2023[6-8].

Animal-to-human transmissions were responsible for early sporadic human Mpox cases, while human-to-human transmission caused the recent MPXV outbreak[1]. Historically, the transmission of MPXV between humans has been thought to occur primarily through respiratory droplets[9,10]. However, close contact or sexual contact with infectious sores or lesions on mucous membranes has been thought to be the primary mode of transmission during the 2022 outbreak[1,10]. Recent studies have found that MPXV can be detected in specimens from multiple sites of Mpox cases, including saliva, rectal swabs, blood, urine and semen[10-17], while skin lesions have been presumed to be the primary source of viral shedding and believed to be optimal for laboratory diagnosis[1]. However, one recent study has found patients negative for MPXV DNA in skin lesions while positive in other sites, highlighting the important role of multi-site sampling in the diagnosis of Mpox[18]. Although some studies have compared the diagnostic accuracy of different sample sites, evaluations with detailed temporal data are currently lacking. Moreover, asymptomatic and atypical MPXV infections, which may serve as important source of transmission have been found[19-22], and the diagnostic accuracy of some alternative samples remains controversial[23], posing challenges to laboratory diagnosis of Mpox. In addtion, current understanding of MPXV shedding dynamics is restricted by either limited numbers of participants and sampling frequency, short follow-up periods or a bias towards Mpox patients with relatively mild disease[10,12,15,24], which merits further investigation. Notably, transmission of MPXV has been shown to occur indirectly through contaminated fomites[1,4,12]. Therefore, systematic evaluation of the risk of this indirect transmission through environmental fomites is of great value to inform the disinfection and prevention strategies. Previous studies have also shown that MPXV DNA can be detected in different environmental fomites of Mpox patients, while application of the existing data has also been hampered by a limited number of samples and rooms, and lacking the longitudinal sampling in association with disease progression.

Currently, there is no authorized Mpox-specific vaccine available[25]. Vaccines against smallpox have been known to possess cross-protective activity against Mpox, and the Modified Vaccinia Ankara-Bavarian Nordic (MVA-BN, also known as JYNNEOS) has been approved by the US Food and Drug Administration (FDA) for pre-exposure vaccination against Mpox[4,26]. One recent study has estimated that the vaccine effectiveness of MVA-BN against Mpox was 87% (95% confidence interval (CI): 84−90%) for one-dose vaccination and 89% (95% CI: 78−100%) for two-dose vaccination, based on pooled estimates during the 2022 Mpox outbreak[27]. Moreover, low levels of MPXV-neutralizing antibodies were found after MVA-BN vaccination in healthy individuals[28]. Accordingly, it is an urgent need to develop Mpox specific vaccines able to effectively limit Mpox. Similar to other orthopoxviruses, MPXV has two disease-inducing infectious forms, namely the intracellular mature virion (IMV) and the extracellular enveloped virion (EEV), containing six surface proteins, which were speculated to be the main neutralizing antibody-eliciting antigens[29]. However, little is known about the characteristics of surface protein-specific IgG responses following infection and the protective antigens, which is key for the design and development of MPXV-specific vaccines.

In this prospective cohort study, we systematically analyzed the longitudinal positivity rate, viral shedding dynamics of different sample types and environmental fomites from acute MPXV infections in association with disease progression, and the antibody response characteristics against the surface proteins of MPXV.

## Results

### Baseline characteristics of the cohort

During June 11, 2023, and November 13, 2023, a total of 139 laboratory-confirmed Mpox patients were found in Shenzhen, China (Figure S2), and all the patients were men. Totally, 77 out of 81 hospitalized patients were enrolled in this study with a median age of 30 years (Table 1), and only 5 patients received smallpox vaccination during childhood. Of note, 72 patients were men who have sex with men (MSM), and the rest bisexual. Moreover, among the 77 patients, 42 patients were HIV-positive with a median CD4 count of 450 (interquartile range (IQR): 237-566), and the rest were immunocompetent. Skin lesions were the most common initial symptoms with 98.7% (76/77) positivity rate, followed by some influenza-like symptoms, including fever (71.43%), sore throat (61.04%), and myalgia (35.06%). Lymphadenopathy and rectitis were found in 41.56% and 19.48% of the patients, respectively. There were no significant differences of the initial symptoms between HIV and Non-HIV participants, except a significantly higher prevalence of rectitis in the HIV participants (Table 1). The median days between symptom onset to admission is 5, and the median days of hospitalization is 7. The fever clinic was the most common (66.23%) medical setting at first visit, followed by the Department of Dermatology (9.09%), HIV clinic (7.79%), Department of Emergency (6.49%) and Anorectal surgery clinic (5.19%). The most common location of skin lesions were the trunk and extremities (77.92%), genitals (62.34%), face (50.65%) and perianal region (37.66%), and to a lesser extent the hands and feet (12.99%). No participants received specific antiviral treatment for MPXV infection. MPXV genomes were sequenced from all 77 participants, and all MPXV belonged to a newly defined lineage C.1 in the West African clade like other MPXVs circulating in China currently[30].

Altogether, a total of 993 specimens from these patients were serially collected during hospitalization, including 166 oropharyngeal swabs, 154 saliva, 162 rectal swabs, 173 skin lesions, 156 urine and 182 plasma samples (Table 1). Moreover, a total of 1633 environmental fomite swabs from 49 patients were also serially collected, including 89-floor swabs, 125 call button swabs, 124 light switch swabs, 94 television remote control swabs, 124-bed handrail swabs, 125 bedside cupboard swabs, 107 chair (armrest) swabs, 93 door handle (patient room to bathroom) swabs, 93 deposition area (air conditioning air outlet) swabs, 125 mobile phone swabs, 125 clothes swabs, 125 pillow swabs, 94 toilet flush handle swabs, 95 shower handle swabs and 95 delivery window swabs (Table S2). These specimens were further stratified into 3 groups based on the collection time, including the 1 - 7 d.p.o, 8 - 14 d.p.o and 15 - 21 d.p.o groups (Table 1 and S2).

### Positivity rate and dynamics of MPXV in multiple sites of Mpox patients

Overall, 100.00% (76/76), 88.16% (67/76), 83.78% (62/74), 78.95% (60/76), 55.26% (42/76) and 31.17% (24/76) participants showed positive detection of MPXV DNA in skin lesions, rectal swab, saliva, oropharyngeal swab, urine and plasma samples (Table S1) during the follow-up, with no differences between the HIV and Non-HIV groups. For the three groups based on the collection date, all the skin lesions showed the highest positivity rates, followed by rectal swab, saliva, oropharyngeal swab, urine and plasma (Table S1). Of note, positivity rates of 91.67%, 58.33% and 56.52% were found from the skin lesions, rectal swabs and saliva samples in the 15 - 21 d.p.o group, respectively, while oropharyngeal swabs only showed a positivity rate of 25% (Table S1). Moreover, only 2 plasma samples were positive for MPXV DNA in the 15 - 21 d.p.o group. Furthermore, we have also analyzed the positivity rate of 143 paired samples of skin lesions, rectal swabs, saliva and oropharyngeal swabs. Generally, 139, 103, 101, and 70 positive samples were found for skin lesions, rectal swabs, saliva and oropharyngeal swabs, respectively (Fig. 1A). In the 1 - 7 d.p.o group, the positivity rates were similar among rectal swabs, saliva and oropharyngeal swabs

**Table 1 | Epidemiological and clinical features of Mpox patients in this study**

| Characteristics | Total (N = 77) | HIV (N = 42) | Non-HIV (N = 35) | p values[c] (HIV vs Non-HIV) |
|---|---|---|---|---|
| Median age (range) | 30 (21–51) | 31.5 (21–50) | 29 (21–51) | 0.0844 |
| Male (%) | 100.00 | 100.00 | 100.00 | >0.9999 |
| Smallpox vaccination (N, %) | 5 (6.49) | 4 (9.52) | 1 (2.86) | 0.3691 |
| Median CD4 count (IQR) | NA | 450 (237–566) | NA | NA |
| Sexual orientation | | | | |
| MSM | 72 (93.51) | 41 (97.62) | 31 (88.57) | 0.1708 |
| Bisexual men | 5 (6.49) | 1 (2.38) | 4 (11.43) | 0.1708 |
| Initial symptoms (N, %) | | | | |
| Fever | 55 (71.43) | 30 (71.43) | 25 (71.43) | >0.9999 |
| Skin lesions | 76 (98.70) | 41 (97.62) | 35 (100.00) | >0.9999 |
| Exhaustion | 23 (29.87) | 14 (33.33) | 9 (25.71) | 0.6178 |
| Lymphadenopathy | 32 (41.56) | 15 (35.71) | 17 (48.57) | 0.3532 |
| Headache | 11 (14.29) | 6 (14.29) | 5 (14.29) | >0.9999 |
| Myalgia | 27 (35.06) | 17 (40.48) | 10 (28.57) | 0.3406 |
| Rectitis[b] | 15 (19.48) | 12 (28.57) | 3 (8.57) | 0.0417 |
| Sore throat | 47 (61.04) | 25 (59.52) | 22 (62.86) | 0.8175 |
| Cough | 5 (6.49) | 3 (7.14) | 2 (5.71) | >0.9999 |
| Lesion location (N, %) | | | | |
| Genital | 48 (62.34) | 23 (54.76) | 25 (71.43) | 0.1607 |
| Perianal | 29 (37.66) | 17 (40.48) | 12 (34.29) | 0.6411 |
| Face | 39 (50.65) | 23 (54.76) | 16 (45.71) | 0.4959 |
| Hands and feet | 10 (12.99) | 4 (9.52) | 6 (17.14) | 0.4979 |
| Trunk and extremities | 60 (77.92) | 32 (76.19) | 28 (80.00) | 0.7861 |
| Medical setting at first visit (N, %) | | | | |
| Fever clinic | 51 (66.23) | 25 (59.52) | 26 (74.29) | 0.2280 |
| Department of Dermatology | 7 (9.09) | 5 (11.90) | 2 (5.71) | 0.4454 |
| HIV clinic | 6 (7.79) | 6 (14.29) | 0 (0.00) | 0.0290 |
| Anorectal surgery clinic | 4 (5.19) | 3 (7.14) | 1 (2.86) | 0.6214 |
| Department of oncology | 1 (1.30) | 1 (2.38) | 0 (0.00) | >0.9999 |
| Department of Infectious diseases | 3 (3.90) | 1 (2.38) | 2 (5.71) | 0.5880 |
| Department of Emergency | 5 (6.49) | 1 (2.38) | 4 (11.43) | 0.1708 |
| Median days between onset to admission (IQR) | 5 (3–7) | 4.5 (3–7) | 5 (3.5–7.5) | 0.1998 |
| Median days of hospitalization (IQR) | 7 (5–10) | 8 (5–10) | 7 (3.5–9) | 0.8587 |
| Sample types (N) | 993 | 584 | 409 | |
| Oropharynx (N, %) | 166 (16.72) | 98 (16.78) | 68 (16.63) | >0.9999 |
| Saliva (N, %) | 154 (15.51) | 88 (15.07) | 66 (16.14) | 0.6569 |
| Rectum (N, %) | 162 (16.31) | 95 (16.27) | 67 (16.38) | >0.9999 |
| Skin lesions[a] (N, %) | 173 (17.42) | 107 (18.32) | 66 (16.14) | 0.3959 |
| Urine (N, %) | 156 (15.71) | 91 (15.58) | 65 (15.89) | 0.9295 |
| Plasma (N, %) | 182 (18.33) | 105 (17.98) | 77 (18.83) | 0.7396 |
| Number of specimens collected (N, %) | | | | |
| 1~7 d.p.o | 375 (37.76) | 229 (39.21) | 146 (35.70) | 0.2873 |
| 8~14 d.p.o | 464 (46.73) | 262 (44.86) | 202 (49.39) | 0.1748 |
| 15~21 d.p.o | 154 (15.51) | 93 (15.92) | 61 (14.91) | 0.7218 |

[a]Including swabs of lesion surface and exudate.
[b]Includes anal pain, bleeding, and diarrhoea.
[c]Statistical significance was measured using Chi-squared and Fisher's exact tests.
*IQR* Interquartile range.
*MSM* Men who have sex with men.
*NA* Not available.
d.p.o: Days post symptoms onset.

(Fig. 1B). However, the positivity rates of rectal and saliva swabs were higher than oropharyngeal swabs both in the 8~14 d.p.o (Fig. 1C) and 15~21 d.p.o groups (Fig. 1D). The viral loads (indicated as $log_{10}$ copies per mL) were also analyzed, and the highest viral loads were found in the skin lesions, followed by rectal swabs, saliva and oropharyngeal swabs (Fig. 1 and Table S1). Moreover, there are no statistical differences for positivity rate and viral loads between HIV and Non-HIV groups, except a significantly higher positivity rate of the urine during 15~21 d.p.o in the Non-HIV group and significantly higher viral loads in the rectal swab in the HIV group (Table S1).

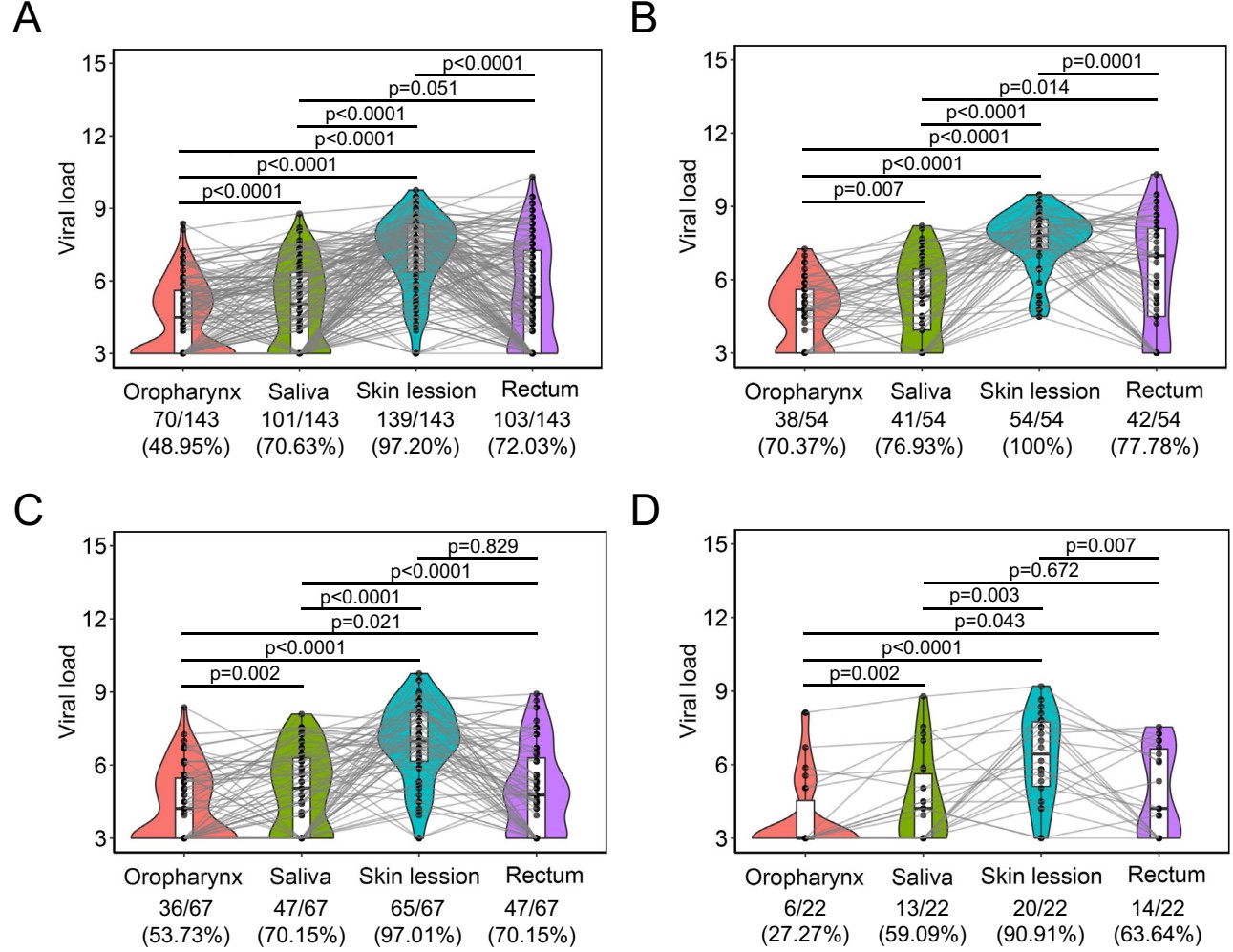

**Fig. 1 | Comparative positivity rates and viral loads among paired samples of skin lesions, rectal swab, saliva and oropharyngeal swabs in association with disease progression. A** The overall comparative positivity rates and viral loads among paired samples of skin lesions, rectal swab, saliva and oropharyngeal swabs. B-D: Comparative positivity rates and viral loads among paired samples of skin lesions, rectal swab, saliva and oropharyngeal swabs during 1 - 7 d.p.o (**B**), 8 - 14 d.p.o (**C**) and 15 - 21 d.p.o (**D**). Viral load is expressed as $\log_{10}$ copies per mL.

Numbers above the violin plot denote the positivity rate of samples. Viral load between different sample types were analyzed using two-tailed Wilcoxon signed-rank test, and $p$ values less than 0.05 were considered statistically significant. The box plots within the violin plots contain the first quartile (the 25th percentile, minima), the third quartile (the 75th percentile, maxima) and the median (the solid line in the box plots) of the viral loads.

Then we analyzed the viral dynamics in different types of specimens from men with MPXV infection (Fig. 2 and Figure S3). For the skin lesions, viral loads slightly increased and reached the peak at about 6 d.p.o, and then slightly decreased (Fig. 2A). For the rectal swabs, high viral loads were found at the beginning of symptom onset, and decreased rapidly with disease progression (Fig. 2C). For saliva and oropharyngeal swabs, dynamics were similar to skin lesions, while viral loads were much lower (Figs. 2E and 2G). When comparing between HIV and Non-HIV groups, the dynamics and peak viral loads were similar for skin lesions, saliva, and oropharyngeal swabs (Figs. 2B, 2D, 2F, 2H and Figure S3). Notably, much higher viral loads during the early phase of infection and peak viral loads were found in the rectal swabs from the HIV group (Fig. 2E and Figure S3). Then we further analyzed the impact of rectitis on the viral dynamics of MPXV, and found that Mpox patients with rectitis showed significantly higher peak viral load (Figure S4A) and much higher viral loads during disease progression (Figure S4B). To verify the possible roles of HIV and rectitis in the rectal viral loads, we further compared the viral loads between patients with and without HIV among the participants without rectitis (Figure S4C and S4D), and significantly higher peak viral loads and slightly higher viral loads during disease progression were found in the HIV

group. Moreover, a comparison between patients with and without rectitis among the participants with HIV also showed significantly higher peak viral load and much higher viral loads during disease progression in patients with rectitis (Figure S4E and S4F).

**Persistent detection of MPXV DNA in environmental fomites from Mpox patients**

Totally, 52.66% (860/1633) of environmental fomite swabs were positive for MPXV DNA, with 69.89% (65/93) of the deposition area (Air conditioning air outlet) swabs, 68% (85/125) of the pillow swabs, 62.92% (56/89) of the floor swabs, 61.6% (77/125) of the bedside cupboard swabs, 59.68% (74/124) of the bed handrail swabs, 56.8% (71/125) of the clothes swabs, 54.2% (58/107) chair (arm rest) swabs, 52.8% (66/125) of the mobile phone swabs, 47.37% (45/95) of the shower handle swabs, 45.74% (43/94) of the toilet flush handle swabs, 45.16% (42/93) of the door handle (patient room to bathroom) swabs, 44.68% (42/94) of the television remote control swabs, 43.2% (54/125) of the call button swabs, 37.89% (36/95) delivery window swabs and 37.1% (46/124) of the light switch swabs, respectively. The mean viral loads of these samples were 5.37 $\log_{10}$ copies per mL (Ct value: 32.83), and significant differences were found among different swabs (ANOVA, $p = 0.008$),

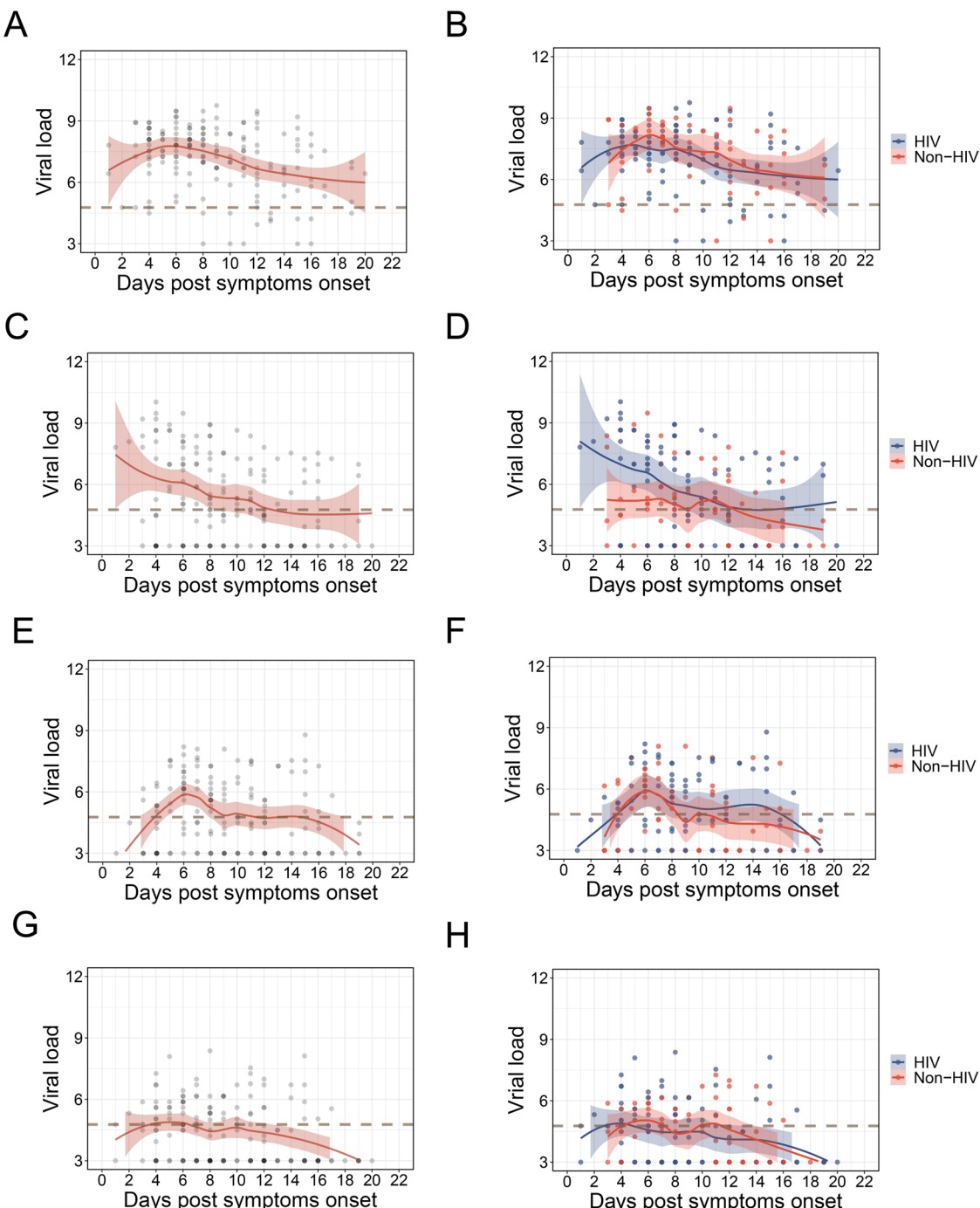

**Fig. 2 | Viral shedding dynamics of MPXV in different sites of Mpox patients. A, C, E** and **G** show the overall viral shedding dynamics of MPXV in skin lesions (**A**), rectal swab (**C**), saliva (**E**) and oropharyngeal swabs (**G**) from Mpox patients. **B, D, F** and **H** show the comparative viral shedding dynamics of MPXV in skin lesions (**B**), rectal swab (**D**), saliva (**F**) and oropharyngeal swabs (H) between Mpox patients with (HIV) or without HIV (Non-HIV). Viral load is expressed as $\log_{10}$ copies per mL.

The kinetics of viral loads during hospitalization were calculated by the LOESS (locally estimated scatterplot smoothing) curve fitting polynomial regression using R. The solid lines represent the fitted curve obtained using the LOESS curve fitting polynomial regression, and the gray band areas represent 95% confidence intervals. The dotted line means the viral load of 4.77 $\log_{10}$ copies per mL (a Ct value of about 35).

among which mean viral loads in the deposition area (5.82 $\log_{10}$ copies per mL) is the highest (Fig. 3A and Table S3). By further dividing the samples according to their collection time, we found no statistical differences among the 1 - 7 d.p.o, 8 - 14 d.p.o and 15 - 21 d.p.o groups (Table S3). Previous studies have successfully recovered viable MPXV from environmental fomite swabs with the lowest viral load of 6.59 $\log_{10}$ copies per mL ($3.9 \times 10^6$ copies per mL)[31,32], indicating a high risk of transmission when the viral load was higher than that. Therefore, we

then further analyzed the distribution of viral loads from these environmental fomites. The results showed that the proportion of swabs with viral loads higher than 6.59 $\log_{10}$ copies per mL were highest for the deposition area (17.2%), followed by bedside cupboard (11.2%), floor (11.83%), and then the pillow (10.4%) (Fig. 3B). Moreover, the proportion of swabs with viral loads higher than 6.59 $\log_{10}$ copies per mL maintained high and increased with the disease progression for most environmental fomites (Fig. 3C).

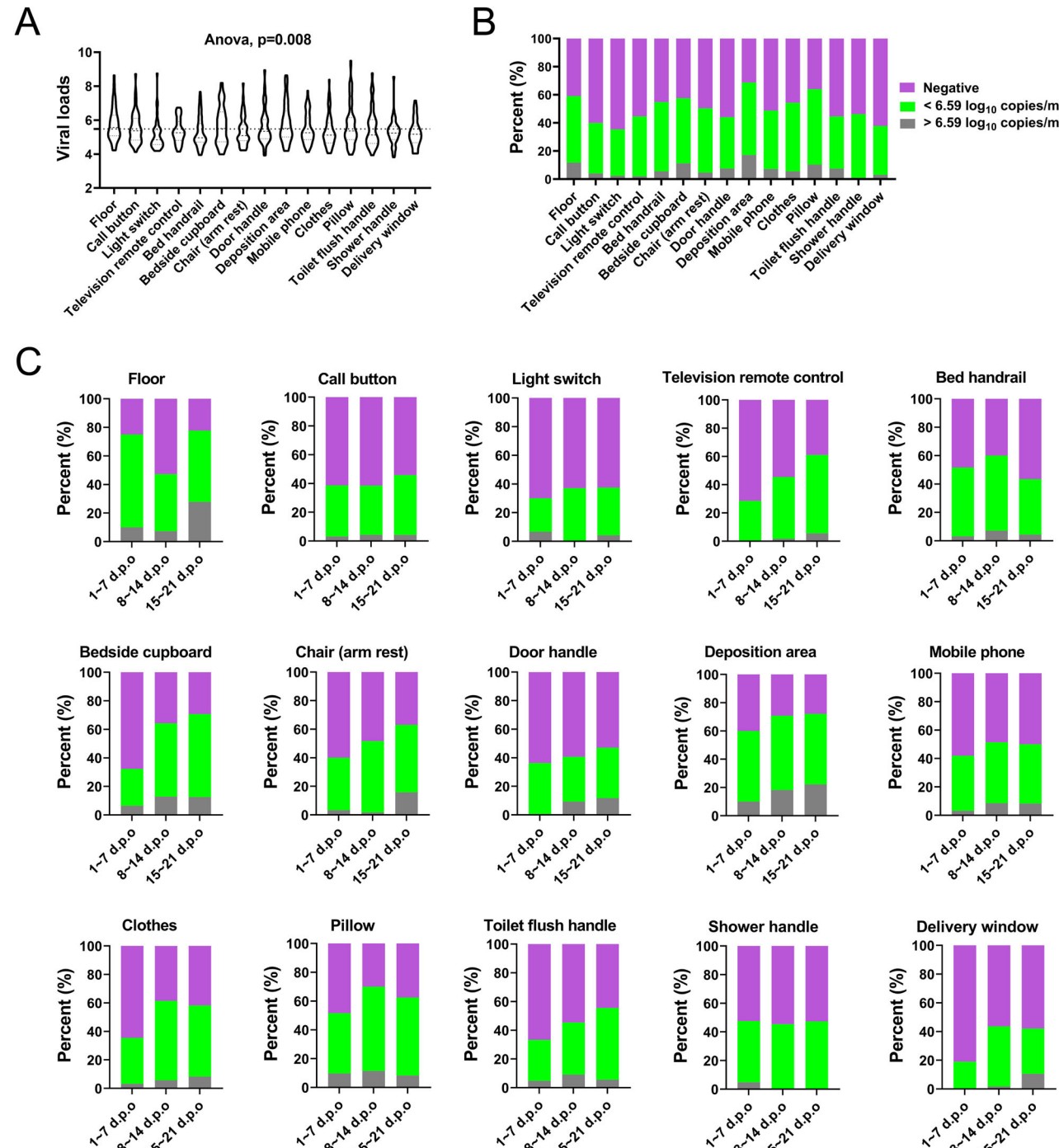

**Fig. 3 | Distribution of the MPXV viral loads on different environmental fomits from Mpox patients in association with disease progression. A** Comparison of the MPXV viral loads on different environmental fomits from Mpox patients. Black dotted line indicates the mean viral loads on different environmental fomits. The lines within the violin plot indicate the mean and standard deviation of the viral loads. ANOVA was used to compare the viral roads among different environmental fomits. **B** Overall distribution of the MPXV viral loads on different environmental fomits from Mpox patients. **C** Distribution of the MPXV viral loads on different environmental fomits from Mpox patients in association with disease progression.

## Antibody characteristics of men with acute MPXV infection

Plasma samples were also serially collected from these 67 participants and subjected for the test of MPXV-specific IgG antibody response, including A29L, A35R, B6R, E8L, H3L and M1R proteins. About 94.02%, 16.42%, 41.79%, 35.82%, 80.6% and 2.99% individuals showed positive IgG response against A29L, A35R, B6R, E8L, H3L and M1R, respectively, with no differences between HIV and Non-HIV groups (Fig. 4A). Then we further analyzed the anti-A29L and anti-H3L IgG in detail. The

positivity rate for anti-A29L and anti-H3L IgG during 1~7 d.p.o could reach 86.36 and 79.55, and increased to 100% and 94.74% during 15~21 d.p.o, respectively (Figs. 4B and 4C). The endpoint titers of anti-A29L IgG increased along with the disease progression and reached the peak at about 12 d.p.o, while no increase was found for anti-H3L IgG titers (Fig. 4E). The dynamics of anti-A29L and anti-H3L IgG titers were similar between HIV and Non-HIV groups (Figs. 4D and 4E). Moreover, $ID_{50}$ of 67 plasma samples against authentic MPXV were detected using

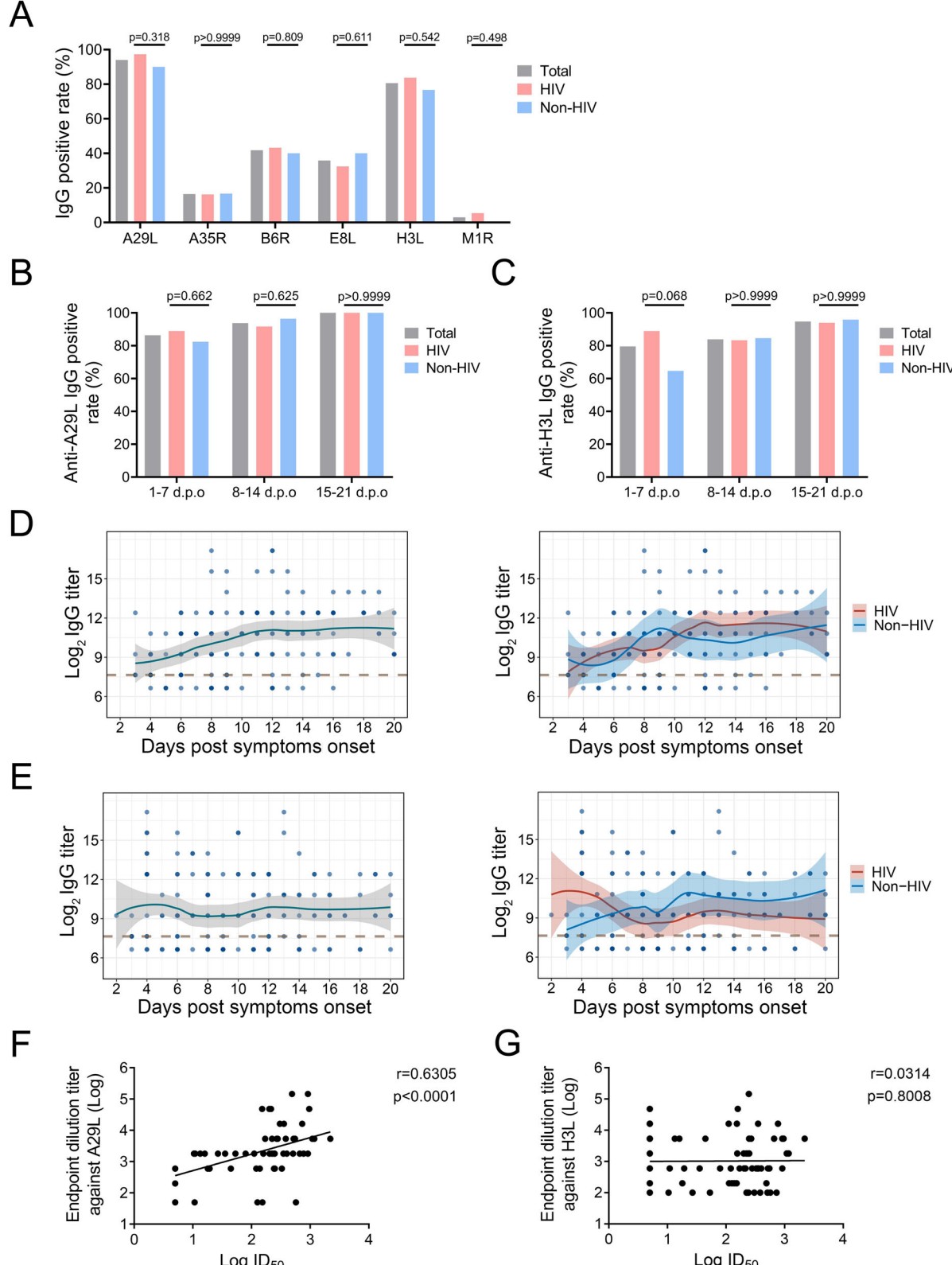

**Fig. 4 | MPXV specific antibody characteristics of acute infections. A** Overall seropositivity of the specific IgG response against different surface proteins of MPXV, including A29L, A35R, B6R, E8L, H3L and M1R proteins. **B** and **C** Comparative seropositivity of the specific IgG response against A29L (**B**) and H3L (**C**) in HIV and Non-HIV participants. Statistical significance was measured using Chi-squared and Fisher's exact tests, and *p* values less than 0.05 were considered statistically significant. **D** and **E** Dynamics A29L (**D**) and H3L (**E**) specific IgG endpoint titers. The antibody kinetics during hospitalization were calculated by the LOESS curve fitting polynomial regression using R. The solid lines represent the fitted curve obtained using the LOESS curve fitting polynomial regression, and the gray band areas represent 95% confidence intervals. **F** and **G** The linear correlation analyses between the A29L (**F**) and H3L (**G**) specific IgG endpoint titers and nAb titers. The dotted line indicates the detection limit of endpoint titers.

FRNT assay, and neutralizing activity was detected in 97.92% (47/48) of the samples collected after 7 d.p.o. The Spearman rank correlation coefficient analyses showed a high correlation between $ID_{50}$ and anti-A29L IgG titer, while not between $ID_{50}$ and anti-H3L IgG titer (Figs. 4F and 4G).

## Discussion

The qRT-PCR assay has been regarded as the gold standard and the most common method for the diagnosis of MPXV infection[23]. Viral shedding patterns varied in different sites of viral infections[10,33], and diagnostic accuracy of different sample types also varied in association with disease progression[34]. In our study, the specimens were further stratified into 1 ~ 7 d.p.o, 8 ~ 14 d.p.o and 15 ~ 21 d.p.o groups based on the collection time, covering different stages of acute viral infection. Consistent with previous studies[10,12,17], skin lesions possessed the highest sensitivity for the diagnosis of MPXV infection during the whole infection course, followed by rectal swabs, saliva and then the oropharyngeal swab during disease progression (Table S1 and Fig. 1). It is worth noting that the positivity rate of oropharyngeal swabs is similar to that of rectal swabs and saliva during 1 ~ 7 d.p.o, while it decreased rapidly and showed much lower positivity rate between 8 ~ 21 d.p.o, which might explain the controversial results about the diagnostic accuracy of oropharyngeal swabs[23]. The positivity rates of rectal swabs and saliva maintain similar during disease progression and are close to the positivity rates of skin lesions, indicating that these two types of specimens could serve as excellent alternatives for the diagnosis of MPXV infection, especially for asymptomatic and atypical infections when skin lesions are not available. In addition, serological assessment of MPXV infection could be a complementary method for the diagnosis of Mpox, and useful in some other settings, including identification of self-attenuated infection, and assessing population seroprevalence to determine asymptomatics[35]. Consistent with a previous study with the whole virus as the coated antigen[36], MPXV-specific antibodies could be detected within 7 d.p.o and maintained stable at 20 d.p.o. Among the six main neutralizing-eliciting antigens of MPXV, much higher seropositivity rates of A29L and H3L specific IgG were found with over 80% seropositivity rates during the first week in our study (Fig. 4), indicating two prominent serological markers for the early diagnosis of MPXV infection. Similar to a previous study, the lowest seropositivity of M1R (homologous protein of L1R) specific IgG was found in our study, suggesting that MPXV infection fails to induce sufficient M1R-specific antibodies[37].

Human-to-human transmission of MPXV has been shown to occur through direct contact with skin lesions, body fluids, or respiratory droplets from infected humans[1,9,12], therefore, viral load and viral shedding dynamics of different sample types have important implications on transmissibility, treatment and public policy[38]. Consistent with previous studies, our results showed that skin lesions possessed the highest viral loads, followed by anal swabs, saliva, and oropharyngeal swabs, and the viral loads in urine and plasma were low[12]. These high positivity rates and viral loads in skin lesions and rectal swabs are in line with previous findings that direct contact with infectious sores or lesions on mucous membranes has been the primary transmission mode of transmission during the 2022 outbreak[1,39–43]. Moreover, the high occurrence of genital and perianal lesions further support that close contact during sex might be the dominant form of transmission in the current outbreak as previously reported[42]. Meanwhile, the high positivity rates of MPXV in saliva and oropharyngeal swabs highlight the transmission of MPXV through breathing and kissing as previously reported[12,15]. Although the association between cell culture positivity and viral loads (both Ct values and copies per ml) could be influenced by qRT-PCR assays, sample storage and processing, as well as cell culture methods, previous studies have suggested that a Ct value of ≥ 35 (≤4.77 $\log_{10}$ copies per mL in our study) corresponds with non or marginal infectivity[14,44–47]. Our viral kinetic analyses revealed that the viral load in skin lesions peaked at around day 6 d.p.o, followed by a gradual decline, with the viral loads still above 4.77 $\log_{10}$ copies per mL at 21 d.p.o. Similar kinetics were found for the saliva, and oropharyngeal swabs, with much lower viral loads (peak viral loads were slightly higher than 4.77 $\log_{10}$ copies per mL for saliva and around 4.77 $\log_{10}$ copies per mL for oropharyngeal swabs) when compared with skin lesion (Fig. 1, Fig. 2 and Figure S3). Moreover, our results from rectal swabs showed the highest viral loads following symptom onset, and then rapidly decreased to below 4.77 $\log_{10}$ copies per mL at approximately 12 d.p.o (Fig. 2C). These results indicate that the highest risk of transmission through skin lesion, saliva and breathing occurred at about 6 d.p.o, and immediately post symptom onset for contact with the rectal mucosa. Notably, Mpox patients with HIV (Fig. 2D and Figure S2B) and rectitis (Figure S4) showed significantly higher peak viral load and much higher viral loads during disease progression. Meanwhile, previous studies have shown that the rectal mucosa immune environment in MSM could lead to the recruitment of immune cells[48], and MPXV may spread with the infection of nearby tissue-resident immune cells, potentially including antigen-presenting cells such as monocytes, macrophages, B cells and dendritic cells[4,42]. The high viral loads in the rectum, lower degrees of keratinisation of anorectal and genital epithelium and higher frequencies of antigen-presenting cells in rectal mucosa further highlight the high probability of sexual transmission of MPXV, especially the MSM with HIV and rectitis.

Environmental contamination in the living spaces of Mpox patients have been confirmed and viable MPXV has also been successfully recovered, suggesting a risk of onward transmission to close contacts or members of the general public present in these locations[49,50]. In our study, we collected a total of 1633 environmental fomite swabs from 49 patients covering different stages of acute infection. Our results further confirmed the extensive environmental contamination during disease progression, with no statistical differences of viral loads among the 1 ~ 7 d.p.o, 8 ~ 14 d.p.o and 15 ~ 21 d.p.o groups (Table S3). Notably, a substantial proportion of samples with viral load > 6.59 $\log_{10}$ copies per mL were found, with the highest for deposition area, followed by floor and bedside cupboard (Fig. 1B). Moreover, the proportion of samples with viral load > 6.59 $\log_{10}$ copies per mL were higher in 8 ~ 14 d.p.o and 15 ~ 21 d.p.o groups in most cases (Fig. 1C). These results suggest the potentially high and persistent prevalence of infection-competent MPXV around living spaces of Mpox patients and the high risk of transmission to close contacts. Although our study was based on the specialist healthcare environment, results may be widely applicable to other spaces and settings where Mpox patients stay for prolonged periods. Therefore, surface cleaning protocols, the use of appropriate personal protective equipment (PPE), and robust doffing procedures are of great importance to avoid potential onward transmission to family members and healthcare staff[49,51].

Researchers are developing mRNA and sub-unit vaccines based on different sets of antigens[52–59]. Several surface proteins of MPXV, including M1R, E8L, H3L, A29L on the IMV, A35R, and B6R proteins on the EEV, have been found to be potential vaccine targets for MPVX[29]. However, which protein could serve as the main antigen to induce strong neutralizing immunity against MPXV remained controversial in these studies, which might associate with the differences in the design, modification, and composition of antigens, the experimental settings, and the used animal models[60]. Moreover, nearly all the evaluation of the neutralizing and protection efficiency were done with VACV, while the intrinsic differences between VACV and MPXV might influence the application of obtained results. Consistent with a previous study, high positivity of neutralizing antibodies were found in Mpox patients[61]. However, in contrast to one recent study, which has found that only A35R-specific IgG titers were associated with the neutralizing titers[62], our results showed the high correlation between the titers of A29L-

specific IgG and the neutralizing titers (Fig. 4G). The discrepancy may be due to the following reasons: Firstly, the neutralization assays conducted by Moraes-Cardoso et al. were done with EEV of MPXV, while a mixture of EEV and IMV was used in our study. Secondly, the clinical samples tested in our study ranged from 3 to 21 d.p.o., while the samples used by Moraes Cardoso et al. were all collected after 29 d.p.o. Moreover, the similar seropositivity rates of the surface proteins between HIV and Non-HIV participants found in our study provide valuable information on the immunogenicity of MPXV antigens in immunocompromised populations.

Previously, all MPXV infections have been thought to be symptomatic and serve as the main source of transmission through close contact[20,63]. The secondary attack rate (SAR) of MPXV was estimated to be 10% for unvaccinated household contacts from Mpox cases with the Congo Basin monkeypox clade, and even lower for the Western African clade responsible for the current endemic[64]. Therefore, outbreaks of Mpox in the general population can potentially be contained in the absence of repeated animal-to-human transmission, evidenced by several previously reported outbreaks[65–67]. Similarly, the first confirmed Mpox case in Shenzhen was found on June 11, 2023[7], followed by a small wave of MPXV infection until August 10, 2023, and later sporadic cases (Figure S2). Fever clinic was the most common (66.23%) medical setting at first visit, and not all cases may have been recognized as it Mpox, particularly if doctors lack relevant experience. Cases with initial influenza-like symptoms were found in our cohort and other studies[22]. Moreover, etiological and serological evidence of asymptomatic MPXV infection and individuals without recognized symptoms have been reported[20,21]. Although asymptomatic carriers were thought to play a negligible role in the spread of orthopox viruses, undiagnosed MPXV infections may play a much more significant role in the transmission of MPXV due to the dense sexual network of MSM men[20], as studies have shown that the basic reproduction number (R0) for MPXV greatly exceeds 1 among networks of MSM compared to R0 levels below 1 in non-MSM sexual networks[68,69].

There are limitations about our study. First, we only detected the DNA of MPXV, and virus isolation assays were not undertaken during the analyses. Second, all the enrolled patients discharged within 21 d.p.o., and viral clearance were not observed for most of them. Therefore, we could not analyze the time for viral clearance. Third, our results suggested a possible impact of rectitis on the viral dynamics of the rectum in the Mpox patients, but the patient number with rectitis was small. This observation merits further investigation in a large cohort.

In conclusion, our results contribute to an improved understanding of the sample selection for higher laboratory diagnosis accuracy at different stages of infection, the viral shedding kinetics of multiple sites from Mpox patients, the transmission risk of fomites during disease progression, and the antibody response characteristics against the main neutralizing antibody-eliciting antigens of MPXV. With the ongoing outbreak of Mpox worldwide, these data could provide useful information for the diagnosis, treatment, and prevention of transmission, and the development of vaccines for Mpox.

## Methods

### Study design and participants
We did a prospective cohort study on the longitudinal viral shedding and antibody response characteristics from men with acute infection of MPXV admitted to Shenzhen Third People's Hospital during June 9, 2023, to November, 5, 2023 ($N = 77$) (Shenzhen, Guangdong, China) for isolation and treatment, and all the patients were followed-up to discharge. All the patients were living in the negative pressure isolation ward during hospitalization to reduce the possibility of further transmission. Clinical and epidemiological information were collected at the earliest time-point after admission. Each Ct value of quantitative real-time PCR (qRT-PCR) from different sample types from the patients

and the environmental fomites, and the specific IgG antibody against different MPXV surface proteins (including A29L, A35R, B6R, E8L, H3L and M1R) during hospitalization were detected and collected. For analyses, the participants were grouped based on the status of HIV infection (HIV, $N = 42$ and Non-HIV, $N = 35$). Moreover, all these specimens were further stratified into 3 groups based on the collection time related to the days post symptoms onset (d.p.o), including the 1 - 7 d.p.o, 8 - 14 d.p.o and 15 - 21 d.p.o groups. The study protocol was approved by the Ethics Committees of Shenzhen Third People's Hospital (2021-030), and written informed consents were obtained.

### Sample collection and detection of the MPXV DNA
Serial swabs from oropharynx, skin lessions and rectum, and samples of saliva (about 0.3–0.5 ml), urine (3–5 ml) and plasma (2–3 ml) were collected by professional nurse every two to three days. Samples from the fomits were also serially collected by the staffs in the Department of Healthcare-associated Infection Management every two days before the disinfection and clean. After the sample collection, the rooms were cleaned every 24 hours using 5000 ppm available chlorine sodium hypochlorite on all hard surfaces and floors and 10000 ppm available chlorine sodium hypochlorite for the toilet, shower, and washbasins as previously reported49. All the swabs and saliva samples were dissolved with about 2 mL of viral transport medium, and the nucleic acid was extracted according to the manufacturer's instructions using 200 µl sample (Daan Co., Ltd.). Then the samples were subjected for the detection of MPXV DNAs using commercial kits based on qRT-PCR following the manufacturer's instruction (GeneoDX Co., Ltd.). The specimens were considered positive if the Ct value was <40, and negative if the results were undetermined. The copy number per mL was determined based on the standard curve generated using the tenfold serially diluted standard plasmid as previously reported (Figure S1)[70]. Viral load measures are expressed as $\log_{10}$ copies per mL (mL of viral transport media for swabs, saliva and skin lessions, and mL of urine and plasma). The sequences of infected MPXV were determined by nanopore sequencing (BGI Co., Ltd.) and the genotypes were analyzed using the Nextclade (https://clades.nextstrain.org/).

### Enzyme-linked immunosorbent assays (ELISAs)
The MPXV surface proteins specific IgG antibodies in plasma specimens were detected using ELISA as previously reported and run in triplicate[59]. Briefly, microtiter plates (Sangon Biotech) were coated overnight at 4 °C with 100 ng of each recombinant protein of MPXV, including A29L, H3L, B6R, A35R, E8L and M1R, which was expressed in E. coli expression system and purified by Ni-NTA. The plates were then washed twice with 1 × PBS (phosphate buffer saline) containing 0.1% v/v Tween-20 (PBST) and blocked with blocking solution (1 × PBS containing 5% w/v bovine serum albumin) overnight at 4 °C. On the day of the experiment, the plates were washed once with PBST. The 3-fold serially diluted sera were added to the wells and incubated at 37 °C for 60 min. Then the plates were washed five times using PBST, and 100 µL horseradish peroxidase (HRP)-conjugated goat anti-human IgG antibody solution (Sangon Biotech) was added to each plate, respectively, and incubated at 37 °C for 60 min. Wash each plate with PBST five times, and 100 µL of tetramethylbenzidine (TMB) substrate (Sangon Biotech) was added to each plate at room temperature in the dark and reacted for 15 min. After the reaction, the plate was stopped with a 2 M $H_2SO_4$ solution. The absorbance was measured at 450 nm using a microplate reader. The ELISA titers were determined by endpoint dilution.

### Focus reduction neutralization test (FRNT)
The neutralizing antibody (nAb) titers of the plasma samples against MPXV were detected using FRNT assay with a clinical isolate of MPXV (hMpxV/China/SZ-SZTH41/2023, EPI_ISL_18213375) following the procedure as previously reported with modification[71]. All plasma samples

were heat-inactivated at 56 °C for 30 min before use. Serial dilutions of tested plasma samples were mixed with an equal volume of monkeypox live virus (200 focus forming units) in 96-well microwell plates and incubated at 37 °C for 1 h in the presence of 10% guinea pig serum (Beijing Bersee Technology Co., Ltd) as a source of complement. Mixtures were then transferred to 96-well plates seeded with $1.5 \times 10^4$ Vero E6 cells per well and incubated at 37 °C for 18 h. Then the supernatant was removed, and cells were fixed with 4% paraformaldehyde solution for 30 min, and permeabilized with 0.15% Triton X-100 for 10 min, followed by incubating with HRP-conjugated Vaccinia Virus Polyclonal Antibody (Invitrogen) for 2 h at room temperature. The reactions were developed with KPL TrueBlue Peroxidase substrates (Seracare Life Sciences Inc.). The numbers of MPXV foci were calculated using an EliSpot reader (Cellular Technology Ltd). Neutralizing antibody titers were calculated as a 50% inhibitory dose ($ID_{50}$) expressed as the dilution of plasma that resulted in a 50% reduction of the numbers of MPXV foci compared with virus control. Samples from 67 participants were tested and used for the linear correlation analysis, and the left 10 participants were not included due to the delayed enrollment and the temporary unavailability of the BSL-3 laboratory.

### Statistical analysis

Statistical analyses were done as previously reported[38,72]. The neutralizing titers of $ID_{50}$ were calculated using a five-parameter dose-response curve in GraphPad Prism. For descriptive analysis, data were presented as median (interquartile range, IQR) for continuous parameters and frequency (percentage) for categorical variables. Chi-squared and Fisher's exact tests were used to compare categorical variables, while for continuous variables, the t-test was used for normal data and the Mann-Whitney U test for non-normal data. Two-tailed Wilcoxon signed-rank test was used to compare the viral loads between the paired samples of different types. ANOVA (Analysis of Variance) was used to compare the viral roads among different environmental fomits. The Spearman rank correlation coefficient was used for linear correlation analysis between the ELISA titers and nAb titers. The kinetics of Ct values and IgG levels during hospitalization were calculated by the LOESS (locally estimated scatterplot smoothing) curve fitting polynomial regression using R version 4.1.0. $P$ values less than 0.05 were considered statistically significant.

### Reporting summary

Further information on research design is available in the Nature Portfolio Reporting Summary linked to this article.

## Data availability

The data supporting the findings of this study are available within the paper and the supplementary information files. Source data are provided with this paper.

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

## Acknowledgements

This work was supported by grants National Science and Technology Major Project (2022YFB3207205), Shenzhen Clinical Research Center for Emerging Infectious Diseases (LCYSSQ20220823091203007), Guangdong Science and Technology Plan Project for Construction of High-level Biosafety Laboratories (2021B1212030010), Shenzhen High-level Hospital Construction Fund (23250G1001 and XKJS-CRGRK-004) and Sanming Project of Medicine in Shenzhen (SZSM202311033). The funders had no role in study design, data collection and analysis, decision to publish, or preparation of the manuscript.

## Author contributions

Y.Y., Y.L. and H.L. conceived and designed the study. Y.Y., C.S., L.G. and L.S. contributed to the analysis and interpretation of data. L.Y., F.W., Y.X., S.C. and T.H. enrolled of patients and collection of samples. S.N., S.S., Y.P., S.Z., J.C., Z.L. and Y.C. carried out the experiment. L.Y., J.L. and S.C. collected the clinical data. Y.Y. and S.S. drafted the article. Y.Y., L.Y., S.S., C.S. and S.N. have accessed and verified all the data reported in this study. All authors reviewed and revised the manuscript and approved the final version. The corresponding author attests that all listed authors meet authorship criteria and that no others meeting the criteria have been omitted.

## Competing interests

The authors declare no competing interests.

## Additional information

[1]Shenzhen Key Laboratory of Pathogen and Immunity, State Key Discipline of Infectious Disease, Shenzhen Third People's Hospital, Second Hospital Affiliated to Southern University of Science and Technology, Shenzhen, China. [2]Guangdong Key Laboratory for Diagnosis and Treatment of Emerging Infectious diseases, Shenzhen, China. [3]National Clinical Research Center for Infectious Disease, Shenzhen, China. [4]BSL-3 Laboratory (Guangdong), Guangdong Provincial Key Laboratory of Tropical Disease Research, School of Public Health; Department of Laboratory Medicine, Zhujiang Hospital, Southern Medical University, Guangzhou, China. [5]Department of Central Laboratory, Xiangyang Central Hospital, Affiliated Hospital of Hubei University of Arts and Science, Xiangyang, Hubei, China. [6]School of Public Health, Bengbu Medical College, Bengbu, Anhui, China. [7]These authors contributed equally: Yang Yang, Shiyu Niu, Chenguang Shen, Liuqing Yang, Shuo Song. ✉e-mail: young@mail.sustech.edu.cn; 13927486077@163.com; luhongzhou@szsy.sustech.edu.cn; yingxialiu@hotmail.com

