## [Peer Review File · Nature Communications]

Longitudinal viral shedding and antibody response characteristics of men with acute infection of monkeypox virus: a prospective cohort studyReviewers' Comments:

Reviewer #1:

Remarks to the Author:

Very comprehensive article on the dynamics of antibodies and the virus both with multi-site samples on the patient and in the environment. The patient data on the multisite are not original and correspond to data already published but with more samples. What is interesting is to compare the environment and especially the type of antibodies present in a strategy for improving our vaccination policies.

Reviewer #2:

Remarks to the Author:

- What are the noteworthy results?

Yang and colleagues describe thoroughly the viral load kinetics over the first three weeks of monkeypox virus (MPXV) infection in various parts of the body by PCR-detection of MPXV DNA. They also measure MPXV DNA levels in environmental fomites swabs and describe the dynamics of IgG antibodies against different surface proteins of MPXV.

- Will the work be of significance to the field and related fields? How does it compare to the established literature? If the work is not original, please provide relevant references.

The work is of significance to the mpox field, although not being the first work describing these three items. Previous works had measure MPXV DNA levels in distinct body parts, in environmental fomites and describe the dynamics of antibody responses, several of which have not been referenced in the manuscript (Mpox immune responses: Colavita et al, Journal of Medical Virology 2023; Hubert et al Cell Host Microbe 2023; Cohn et al, Lancet Infect Dis 2023; Moraes-Cardoso et al, Lancet Microbe in press / Lancet preprint.; Mpox fomites: Morgan et al, Emerg Infect Dis 2022).

Despite not being novel, the sound methodology of systematic collection of samples makes the work valuable for the field.

- Does the work support the conclusions and claims, or is additional evidence needed?

Overall, the conclusions and claims expressed by the authors are supported by the data. The authors claim that their data has "profound implications for the diagnosis, treatment, prevention of transmission and development of vaccines for Mpox", which I disagree given the previous knowledge on this field. The lack of viral isolation assays prevents from really translating their findings on MPXV DNA levels into actual risk of virus transmission.

- Are there any flaws in the data analysis, interpretation and conclusions? - Do these prohibit publication or require revision?

I am concerned about the following analysis/interpretations, which should be revised for its publication:

- Key characteristics of the population where this study is conducted are missing, which prevents the appropriate interpretation of the study. The authors should specify whether patients were hospitalized because of the severity of the disease or based on isolation procedures. The immune status of participants, particularly HIV-positive individuals should be specified. 5 patients received smallpox vaccination but it is not clear whether this vaccines were administered during childhood or as part of the 2022 outbreak response, please specify.

- The authors found a higher prevalence of rectitis among HIV-positive individuals. To my knowledge, this is the first study to find this association and, given the small sample size, the authors should discuss whether this finding is truly reflecting an increased risk of rectitis among this population. In the same line, the authors observed an increased rectal viral load in HIV positive individuals. I strongly suggest that the authors conduct a sensitivity analysis comparing rectal viral loads between individuals with and without HIV, and with and without rectitis, to rule out whether the higher rectal viral load in

HIV positive individuals is only reflecting the higher percentage of patients with rectitis in this subgroup.

- Samples from distinct body regions are collected differently and processed differently to measure MPXV DNA. Hence, viral loads between certain samples cannot be compared and this should be clearly specified in the manuscript. In the methods section it is not clear how each sample was collected and processed. For example, the volumes of saliva, urine or plasma used; or if swabs were placed in viral transport media, which total volume of media was present in each tube, and which volume was used for DNA extraction. Viral loads measured in urine and plasma are not fully reported and they should be included.

The authors compare the positivity rates and viral loads from skin lesions, rectal swabs, saliva and oropharyngeal swabs. Only if the authors collected saliva with a swab, results from these samples can be presented together. If not, I would separate the results of these samples so that the reader is not misled to compare the results of one and other.

- As previously mentioned, the authors have not included relevant references describing Mpox immune responses. The authors should discuss the differential findings between their study and the published literature in terms of antibody responses and neutralization.

- Is the methodology sound? Does the work meet the expected standards in your field?

The methodology is sound except for the lack of viral isolation assays. It would also be highly valuable that the authors transformed the Ct values in viral copies so that their results were more comparable with the literature. The reason for discussing results above/below a Ct value of 30 should be specified

- Suner and colleagues defined a threshold for culture positivity of $6.5 \log_{10}$ copies per mL or higher (approximately a Ct value of 26).

- Is there enough detail provided in the methods for the work to be reproduced?

As previously stated, the methods section lacks information on the procedures for sample collection and processing.

Additional comments:

- Line 244. This sentence refers to patients' positivity in each body sample at baseline or at any timepoint? Please clarify.

- Line 282 - Results paragraph on environmental fomites. I suggest the authors present these results ordering the locations from higher positivity rates to lower positivity rates.

- Figure 1. Specify whether the statistical analysis compares viral loads or positivity rates. According to the figure, it should compare viral loads, and the Figure title should state "Comparative positive rates and viral loads.."

- Figure 1A. I suggest removing this panel, since panels B-D where data is separated by d.p.o. are more informative. In these panels, I would use the same scale for all y-axes so that they are comparable.

- The authors could potentially explore the correlations between severity and IgG responses, as done by Moraes-Cardoso and colleagues.

Reviewer #3:

Remarks to the Author:

This study predominantly describes the dynamics of viral loads and viral shedding of MPXV infection within a hospital setting. IgG responses to key MPXV antigens are also explored by ELISA (endpoint titres) and FRNT. Structuring the results into HIV status is central as this is an important at-risk group to consider for MPXV infection and transmission currently.

The interpretations of results are clear and sufficiently justified with the following conclusions standing out as most significant:

- Correlation of patient anti-A29 and anti-H3 IgG titres to neutralisation titres. Information on the

contribution of antibodies to particular targets will be crucial for establishing correlates of protection and the next generation of MPXV vaccines.

- Differences in ct values between patient sampling sites over time. This will be important for diagnostic decisions.
- The detailed notes on clinical presentation will aid in accurate diagnosis.
- Of additional note is the comprehensive examination of environmental fomites in a clinical setting. Valuable information for limiting transmission. However, assessing the viability of these samples would have strengthened these results significantly.

The methodology is sound, of a good standard and sufficiently detailed for reproduction. The use of endpoint titres to compare IgG responses is appropriate in the absence of a standard. The two main limitations of the study are also sufficiently highlighted.

Please find my questions and comments below:

- Line 66 (Introduction): Worth expanding on which genetic changes may be responsible.
- Line 168 (Methods: Enzyme-linked immunosorbent assays (ELISAs)): The use of a HRP-conjugated goat anti-human IgM antibody is mentioned in the methods to detect patient IgM to various MPXV antigens. However, the results are not shown or discussed. Suggest either removing this from the methods or presenting the results. The dynamics of IgM responses to these antigens and differences to IgG will be of interest. Their omission does not detract from the conclusions of the study.
- Line 215 (Results: Baseline characteristics of the cohort): This sentence may need rewording to clarify – though Table S1 does this well.
- Line 215 (Results: Baseline characteristics of the cohort): It would be of interest to hear if there were any differences in those vaccinated individuals with regards to viral loads/shedding and serologically. Were these patients vaccinated in response to the current outbreak or previously?
- Line 230 (Results: Baseline characteristics of the cohort): Is this the same lineage as circulating in other cities in China?

Overall, this is a detailed and well-reasoned study which contributes to the field and will be of importance for addressing the current outbreak.

Reviewers' Comments:

Reviewer #1 (Remarks to the Author):

Very comprehensive article on the dynamics of antibodies and the virus both with multi-site samples on the patient and in the environment. The patient data on the multisite are not original and correspond to data already published but with more samples. What is interesting is to compare the environment and especially the type of antibodies present in a strategy for improving our vaccination policies.

Answer: Thanks for your time and positive rating about the quality of this work.

Reviewer #2 (Remarks to the Author):

- What are the noteworthy results?

Yang and colleagues describe thoroughly the viral load kinetics over the first three weeks of monkeypox virus (MPXV) infection in various parts of the body by PCR-detection of MPXV DNA. They also measure MPXV DNA levels in environmental fomites swabs and describe the dynamics of IgG antibodies against different surface proteins of MPXV.

Answer: Thanks for your time and comments on our manuscript.

- Will the work be of significance to the field and related fields? How does it compare to the established literature? If the work is not original, please provide relevant references.

The work is of significance to the mpox field, although not being the first work describing these three items. Previous works had measure MPXV DNA levels in distinct body parts, in environmental fomites and describe the dynamics of antibody responses, several of which have not been referenced in the manuscript (Mpox immune responses: Colavita et al, Journal of Medical Virology 2023; Hubert et al Cell Host Microbe 2023; Cohn et al, Lancet Infect Dis 2023; Moraes-Cardoso et al, Lancet Microbe in press / Lancet preprint.; Mpox fomites: Morgan et al, Emerg Infect Dis 2022). Despite not being novel, the sound methodology of systematic collection of samples makes the work valuable for the field.

Answer: Thanks for your time and positive rating about the quality of this work. We have searched related articles and added some references including those you mentioned in our revised manuscript, and we have added some description and discussion based on the added references (lines 359-367 and lines 438-446).

- Does the work support the conclusions and claims, or is additional evidence needed? Overall, the conclusions and claims expressed by the authors are supported by the data. The authors claim that their data has “profound implications for the diagnosis, treatment, prevention of transmission and development of vaccines for Mpox”, which I disagree given the previous knowledge on this field. The lack of viral isolation assays prevents from really translating their findings on MPXV DNA levels into actual risk of virus transmission.

Answer: Thanks for the insightful comments and suggestions. We have changed the description into “With the ongoing outbreak of Mpox worldwide, these data could provide useful information for the diagnosis, treatment, prevention of transmission and development of vaccines for Mpox”.

- Are there any flaws in the data analysis, interpretation and conclusions? - Do these prohibit publication or require revision?

I am concerned about the following analysis/interpretations, which should be revised for its publication:

- Key characteristics of the population where this study is conducted are missing, which prevents the appropriate interpretation of the study. The authors should specify whether patients were hospitalized because of the severity of the disease or based on isolation procedures. The immune status of participants, particularly HIV-positive individuals should be specified. 5 patients received smallpox vaccination but it is not clear whether this vaccines were administered during childhood or as part of the 2022 outbreak response, please specify.

Answer: Thanks for the insightful comments and suggestions. All the enrolled patients came to visit the doctor actively with clinical symptoms, and most patients (76/77) with skin lesions. Mpox is novel and full of unknown for most Chinese, therefore the laboratory-confirmed patients were advised and got hospitalized voluntarily for both isolation and treatment (lines 124-128). Meanwhile, we have added the description on the immune status of enrolled patients (lines 223-225), and clarified the time-points of vaccination of the 5 patients (lines 220-222).

- The authors found a higher prevalence of rectitis among HIV-positive individuals. To my knowledge, this is the first study to find this association and, given the small

sample size, the authors should discuss whether this finding is truly reflecting an increased risk of rectitis among this population. In the same line, the authors observed an increased rectal viral load in HIV positive individuals. I strongly suggest that the authors conduct a sensitivity analysis comparing rectal viral loads between individuals with and without HIV, and with and without rectitis, to rule out whether the higher rectal viral load in HIV positive individuals is only reflecting the higher percentage of patients with rectitis in this subgroup.

Answer: Thanks for the insightful comments and suggestions. We have transformed the Ct values into viral copies based on the generated standard curve, and we have compared the peak rectal viral loads and viral dynamics between individuals with and without rectitis following your suggestion (Figure S4). Despite of the small sample size, we still found that Mpox patients with rectitis showed significantly higher peak viral load (Figure S4A) and much higher viral loads during disease progression (Figure S4B). These results indeed suggested a possibly significant impact of rectitis on the viral dynamics of rectum in the Mpox patients, which merits further investigation based on a large cohort. Based on the results, we have also added related discussion in the revised manuscript (lines 397-406 and lines 476-478).

- Samples from distinct body regions are collected differently and processed differently to measure MPXV DNA. Hence, viral loads between certain samples cannot be compared and this should be clearly specified in the manuscript. In the methods section it is not clear how each sample was collected and processed. For example, the volumes of saliva, urine or plasma used; or if swabs were placed in viral transport media, which total volume of media was present in each tube, and which volume was used for DNA extraction. Viral loads measured in urine and plasma are not fully reported and they should be included.

Answer: Thanks for the insightful comments and suggestions. We have added the detailed description on the collection and procession of the samples (lines 142-146 and lines 149-152). Actually, serial swabs from oropharynx, skin lesions, rectum and environmental fomits, and samples of saliva (about 0.3-0.5 ml), urine (3-5 ml) and plasma (2-3 ml) were collected. Then, all the swabs and saliva samples were dissolved with about 2 mL of viral transport medium, and 200 µl samples were subjected for the extraction of the nucleic acid.

The authors compare the positivity rates and viral loads from skin lesions, rectal swabs, saliva and oropharyngeal swabs. Only if the authors collected saliva with a swab, results from these samples can be presented together. If not, I would separate the results of these samples so that the reader is not misled to compare the results of one and other.

Answer: Thanks for the insightful comments and suggestions. The saliva samples were collected with a total volume of about 0.3-0.5 ml, and then dissolved with about 2 mL of viral transport medium like the swabs. Indeed, there is a difference of 0.2-0.5 ml between the total sample volumes of saliva and swabs, while very little influence on the positivity rates and viral loads. Therefore, we speculate that they were comparable.

- As previously mentioned, the authors have not included relevant references describing Mpox immune responses. The authors should discuss the differential findings between their study and the published literature in terms of antibody responses and neutralization.

Answer: Thanks for the insightful comments and suggestions. We have added the references and the related discussion following your suggestions (lines 359-367 and lines 438-446).

- Is the methodology sound? Does the work meet the expected standards in your field? The methodology is sound except for the lack of viral isolation assays. It would also be highly valuable that the authors transformed the Ct values in viral copies so that their results were more comparable with the literature. The reason for discussing results above/below a Ct value of 30 should be specified - Suner and colleagues defined a threshold for culture positivity of $6.5 \log_{10}$ copies per mL or higher (approximately a Ct value of 26).

Answer: Thanks for the insightful comments and suggestions. We have generated the standard curve using the ten-fold serially diluted standard plasmid and transformed the Ct values into viral copies. We have carefully searched about the correlation between Ct values/viral loads and viral isolation. Several studies and a meta review have investigated the Ct values/viral loads of clinical samples and viral isolation (Lancet Infect Dis. 2023 Apr;23(4):445-453.; J Hosp Infect. 2023 May;135:139-144.; J Clin Virol. 2023 Apr;161:105421.; Emerg Infect Dis. 2023 Jul;29(7):1465-1469.;

Euro Surveill. 2022 Sep;27(35):2200636.; Euro Surveill. 2022 Sep;27(36):2200675.; J Travel Med. 2023 Sep 5;30(5):taad111.), and found that when the Ct values is above 35 (equivalent to be 4.77 log₁₀ copies per mL in our study). Therefore, we defined a threshold of 4.77 log₁₀ copies per mL to discuss the transmission risk via different sites of Mpox patients (lines 382-397). As to the environmental fomits, previous studies have successfully recovered viable MPXV from environmental fomit swabs with lowest viral load of 6.59 log₁₀ copies per mL (3.9×10^6 copies per mL) (J Hosp Infect. 2023 Jul;137:86-88.; MMWR Morb Mortal Wkly Rep. 2022 Aug 26;71(34):1092-1094.), so we defined a threshold of 6.59 log₁₀ copies per mL to discuss the transmission risk via direct contact with these fomits (lines 309-313 and lines 414-420).

- Is there enough detail provided in the methods for the work to be reproduced?

As previously stated, the methods section lacks information on the procedures for sample collection and processing.

Answer: Thanks for the insightful comments and suggestions. We have added the detailed description on the procedures for sample collection and processing in the methods section following your suggestion (lines 142-146 and lines 149-152).

Additional comments:

- Line 244. This sentence refers to patients' positivity in each body sample at baseline or at any timepoint? Please clarify.

Answer: Thanks for the insightful comments and suggestions. This sentence refers to patients' positivity in each body sample at any time-point. For better understanding, we have added "during the follow-up" at the end of the sentence (line 257).

- Line 282 - Results paragraph on environmental fomites. I suggest the authors present these results ordering the locations from higher positivity rates to lower positivity rates.

Answer: Thanks for the insightful comments and suggestions. We have changed the description following your suggestion (lines 294-303).

- Figure 1. Specificy whether the statistical analysis compares viral loads or positivity rates. According to the figure, it should compare viral loads, and the Figure title

should state “Comparative positive rates and viral loads...”

Answer: Thanks for the insightful comments and suggestions. Actually, the positive rates for different sample types are also shown in the figure above the violin plot.

- Figure 1A. I suggest removing this panel, since panels B-D where data is separated by d.p.o. are more informative. In these panels, I would use the same scale for all y-axes so that they are comparable.

Answer: Thanks for the insightful comments and suggestions. We have unified the scale for all the y-axes following your suggestion, while we have kept the panel of Figure 1A. Actually, Figure 1A could show the readers an overview of the comparative positive rates of multiple sites from the Mpox patients.

- The authors could potentially explore the correlations between severity and IgG responses, as done by Moraes-Cardoso and colleagues.

Answer: Thanks for the insightful comments and suggestions. The analysis on the correlations between severity and IgG responses is useful and informative. However, we are sorry that we did not record the disease severity related index of Mpox, such as the severity score used by Moraes-Cardoso and colleagues. In the future study, we will record the related indexes for the analyses following your suggestion.

Reviewer #3 (Remarks to the Author):

This study predominantly describes the dynamics of viral loads and viral shedding of MPXV infection within a hospital setting. IgG responses to key MPXV antigens are also explored by ELISA (endpoint titres) and FRNT. Structuring the results into HIV status is central as this is an important at-risk group to consider for MPXV infection and transmission currently.

Answer: Thanks for your time and positive rating about the quality of this work.

The interpretations of results are clear and sufficiently justified with the following conclusions standing out as most significant:

- Correlation of patient anti-A29 and anti-H3 IgG titres to neutralisation titres. Information on the contribution of antibodies to particular targets will be crucial for establishing correlates of protection and the next generation of MPXV vaccines.

- Differences in ct values between patient sampling sites over time. This will be important for diagnostic decisions.
- The detailed notes on clinical presentation will aid in accurate diagnosis.
- Of additional note is the comprehensive examination of environmental fomites in a clinical setting. Valuable information for limiting transmission. However, assessing the viability of these samples would have strengthened these results significantly.

Answer: Thanks for your time and positive rating about the quality of this work, and thanks for the insightful comments and suggestions. We strongly agreed with you that assessing the viability of these samples would have strengthened these results significantly. Actually, we have intended to do so at the beginning. However, in terms of bio-safety, most of the samples we received were inactivated, so we could not carry out the viral isolation. Indeed, this is a limitation of our study, and we have described it in the discussion section (lines 472-478).

The methodology is sound, of a good standard and sufficiently detailed for reproduction. The use of endpoint titres to compare IgG responses is appropriate in the absence of a standard. The two main limitations of the study are also sufficiently highlighted.

Answer: Thanks for your time and comments on our work.

Please find my questions and comments below:

- Line 66 (Introduction): Worth expanding on which genetic changes may be responsible.

Answer: Thanks for the insightful comments and suggestions, and we have added the description following your suggestion (lines 64-66).

- Line 168 (Methods: Enzyme-linked immunosorbent assays (ELISAs)): The use of a HRP-conjugated goat anti-human IgM antibody is mentioned in the methods to detect patient IgM to various MPXV antigens. However, the results are not shown or discussed. Suggest either removing this from the methods or presenting the results. The dynamics of IgM responses to these antigens and differences to IgG will be of interest. Their omission does not detract from the conclusions of the study.

Answer: Thanks for the insightful comments and suggestions. We are sorry for the carelessness that we have mistakenly mentioned the IgM, and we have removed it.

- Line 215 (Results: Baseline characteristics of the cohort): This sentence may need rewording to clarify – though Table S1 does this well.

Answer: Thanks for the insightful comments and suggestions. We have revised this sentence for better understanding (lines 220-225).

- Line 215 (Results: Baseline characteristics of the cohort): It would be of interest to hear if there were any differences in those vaccinated individuals with regards to viral loads/shedding and serologically. Were these patients vaccinated in response to the current outbreak or previously?

Answer: Thanks for the insightful comments and suggestions, and we strongly agree with you the mentioned analyses are valuable. Currently, the number of vaccinated individuals in our study is too small (N=5), so we speculate that it is hard to get a conclusive conclusion at this time. Therefore, we will do the analyses in the future based on a larger cohort.

- Line 230 (Results: Baseline characteristics of the cohort): Is this the same lineage as circulating in other cities in China?

Answer: Thanks for the insightful comments and suggestions. Indeed, the circulating lineage of MPXV in China is C.1, and we have added the description (lines 238-240).

Overall, this is a detailed and well-reasoned study which contributes to the field and will be of importance for addressing the current outbreak.

Answer: Thanks for your time and positive rating about the quality of this work.

Reviewers' Comments:

Reviewer #2:

Remarks to the Author:

In my opinion, the manuscript has increased its clarity and its messages are more powerfully delivered, thank you for incorporating most of the suggested changes.

I would still suggest to conduct the comparison of rectal viral loads between patients with HIV and rectitis, with HIV without rectitis, without HIV with rectitis and without HIV without rectitis. This way you will rule out which factors (HIV, rectitis or both) are really mediating an increase in rectal viral loads.

Regarding the figures and tables, I would also suggest some minor changes to facilitate its understanding by readers:

- Table S1 should be main, Table 1 could be supplementary. In fact, data on Table 1 might be easier to read as a supplementary figure.

- In figure 1, I would write the competitive rates below the x axis titles, and also include the rate itself. Example: in graph A, below oropharynx write 49% (70/143)

Congratulations on your work.

Reviewer #3:

Remarks to the Author:

Thank you very much for your rebuttals, all issues have been addressed appropriately. Please find one additional suggestion below:

Line 174 (ELISAs) - please remove mention of the use of goat anti-human IgM antibody if not presenting the data.

Reviewers' Comments:

Reviewer #2 (Remarks to the Author):

1. In my opinion, the manuscript has increased its clarity and its messages are more powerfully delivered, thank you for incorporating most of the suggested changes. I would still suggest to conduct the comparison of rectal viral loads between patients with HIV and rectitis, with HIV without rectitis, without HIV with rectitis and without HIV without rectitis. This way you will rule out which factors (HIV, rectitis or both) are really mediating an increase in rectal viral loads.

Response: Thanks for your time and positive rating about the quality of this work. Following your suggestion, we have conducted the comparison of rectal viral loads between patients with and without HIV among the participants without rectitis (Figure S4C and S4D), and between patients with and without rectitis among the participants with HIV (Figure S4E and S4F), and confirmed that both HIV and rectitis are factors associated with the increase in rectal viral loads.

2. Regarding the figures and tables, I would also suggest some minor changes to facilitate its understanding by readers:

- Table S1 should be main, Table 1 could be supplementary. In fact, data on Table 1 might be easier to read as a supplementary figure.

- In figure 1, I would write the competitive rates below the x axis titles, and also include the rate itself. Example: in graph A, below oropharynx write 49% (70/143)

Congratulations on your work.

Response: Thanks for the insightful comments and suggestions. We strongly agree with your viewpoints about the figure 1 and the tables. We have changed the Table S1 into Table 1 following your suggestion, and added the comparative positive rates below the x axis titles as suggested. Thanks again for your time and the suggestions which greatly improve our work.

3. Reviewer #3 (Remarks to the Author):

Thank you very much for your rebuttals, all issues have been addressed appropriately.

Please find one additional suggestion below:

Line 174 (ELISAs) - please remove mention of the use of goat anti-human IgM antibody if not presenting the data.

Response: Thanks for you time and the insightful comments on our manuscript. We are sorry for the mistake, and we have deleted it in the revised manuscript. Thanks again for your time and the suggestions which greatly improve our work.